# Metabolomic Analyses to Identify Candidate Biomarkers of Cystinosis

**DOI:** 10.3390/ijms24032603

**Published:** 2023-01-30

**Authors:** Emirhan Nemutlu, Fatih Ozaltin, Samiye Yabanoglu-Ciftci, Bora Gulhan, Cemil Can Eylem, İpek Baysal, Elif Damla Gök-Topak, Kezban Ulubayram, Osman Ugur Sezerman, Gulberk Ucar, Sedef Kır, Rezan Topaloglu

**Affiliations:** 1Faculty of Pharmacy, Department of Analytical Chemistry, Hacettepe University, Sihhiye, Ankara 06100, Türkiye; 2Department of Pediatric Nephrology, Hacettepe University School of Medicine, Sihhiye, Ankara 06100, Türkiye; 3Nephrogenetics Laboratory, Department of Pediatric Nephrology, Hacettepe University School of Medicine, Sihhiye, Ankara 06100, Türkiye; 4Faculty of Pharmacy, Department of Biochemistry, Hacettepe University, Sihhiye, Ankara 06100, Türkiye; 5Vocational School of Health Services, Hacettepe University, Ankara 06100, Türkiye; 6Faculty of Pharmacy, Department of Analytical Chemistry, Lokman Hekim University, Sogutozu, Ankara 06510, Türkiye; 7Faculty of Pharmacy, Department of Basic Pharmaceutical Sciences, Hacettepe University, Sihhiye, Ankara 06100, Türkiye; 8Biostatistics and Medical Informatics Program, Faculty of Medicine, Graduate School of Health Sciences, Acibadem Mehmet Ali Aydinlar University, Istanbul 34752, Türkiye

**Keywords:** cystinosis, proteomics, metabolomics, pathway, biomarker, diagnosis, follow-up

## Abstract

Cystinosis is a rare, devastating hereditary disease secondary to recessive *CTNS* gene mutations. The most commonly used diagnostic method is confirmation of an elevated leukocyte cystine level; however, this method is expensive and difficult to perform. This study aimed to identify candidate biomarkers for the diagnosis and follow-up of cystinosis based on multiomics studies. The study included three groups: newly-diagnosed cystinosis patients (patient group, n = 14); cystinosis patients under treatment (treatment group, n = 19); and healthy controls (control group, n = 30). Plasma metabolomics analysis identified 10 metabolites as candidate biomarkers that differed between the patient and control groups [L-serine, taurine, lyxose, 4-trimethylammoniobutanoic acid, orotic acid, glutathione, PE(O-18:1(9Z)/0:0), 2-hydroxyphenyl acetic acid, acetyl-N-formil-5-metoxikinuramine, 3-indoxyl sulphate]. As compared to the healthy control group, in the treatment group, hypotaurine, phosphatidylethanolamine, N-acetyl-d-mannosamine, 3-indolacetic acid, p-cresol, phenylethylamine, 5-aminovaleric acid, glycine, creatinine, and saccharic acid levels were significantly higher, and the metabolites quinic acid, capric acid, lenticin, xanthotoxin, glucose-6-phosphate, taurine, uric acid, glyceric acid, alpha-D-glucosamine phosphate, and serine levels were significantly lower. Urinary metabolomic analysis clearly differentiated the patient group from the control group by means of higher allo-inositol, talose, glucose, 2-hydroxybutiric acid, cystine, pyruvic acid, valine, and phenylalanine levels, and lower metabolite (N-acetyl-L-glutamic acid, 3-aminopropionitrile, ribitol, hydroquinone, glucuronic acid, 3-phosphoglycerate, xanthine, creatinine, and 5-aminovaleric acid) levels in the patient group. Urine metabolites were also found to be significantly different in the treatment group than in the control group. Thus, this study identified candidate biomarkers that could be used for the diagnosis and follow-up of cystinosis.

## 1. Introduction

Cystinosis is a rare disease with an incidence of 0.5–1 per 100,000 individuals [1]. This devastating disease affects all organs; however, kidney impairment—in particular—is most relevant in prognosis. Children with cystinosis appear normal at birth, but present with renal Fanconi syndrome at age 6–9 months, which results from generalized dysfunction of the proximal tubules of the kidneys. For their survival, such patients should take many different drugs to replace urinary losses. Along with this supportive treatment, cysteamine is the specific therapy. Early initiation of cysteamine has been shown to delay progression to chronic kidney failure [1,2]. 

Cystinosis is characterized by the accumulation of cystine crystals in the lysosomes due to a defective cystine/proton co-transporter, cystinosin, which is encoded by the *CTNS* gene and normally functions to transport intralysosomal cystine to the cytoplasm [3]. This accumulation results in impairment of normal cellular functions that eventually leads to multi-organ dysfunction, as CTNS is expressed in all tissues in the human body [4,5]; however, the pathogenesis of cystinosis is not limited to cystine accumulation and is yet to be fully elucidated. Its pathogenesis is complex, as in addition to intralysosomal cystine accumulation, many intracellular pathways, such as inflammatory and fibrotic pathways, autophagy, mTOR signaling, lysosomal biogenesis, and vesicle trafficking, are involved [6,7,8,9,10,11,12]. 

Despite significant advances in our understanding of the pathogenesis of cystinosis, measurement of the leukocyte cystine level (preferentially in granulocytes) remains the current method for diagnosis and monitoring treatment; however, this method has many disadvantages. As granulocytes are short-lived cells, such measurement reflects therapeutic control during only a short duration of time. Moreover, cystine measurement is very sensitive to storage and transport conditions [13]. Genetic testing is an excellent diagnostic tool; however, it provides no benefit to monitoring treatment. Inflammatory markers of macrophage activation, such as IL-1β, IL-6, IL18, and chitotriosidase, have also been considered as potential biomarkers for monitoring treatment [9,14], but these inflammatory markers are not specific to cystinosis; therefore, the development of alternative selective diagnostic and treatment monitoring strategies is essential to advance patient care. 

Living organisms are an integrated and interacting network of genes, transcripts, proteins, small signaling molecules, and metabolites that define cellular phenotype and function. Most of the diseases to be investigated are complicated and a disease’s phenotype is the product of variations in genes, proteins, and metabolites; therefore, integrated “omics” methodologies (i.e., genomics, transcriptomics, proteomics, metabolomics, and fluxomic) represent a viable direction for uncovering alterations in metabolic networks, phenotypes, disease mechanisms, and drug effects [15]. This approach is useful for the identification of novel pathways as well. Recently, Bellomo et al. studied transcriptomic analyses in cystinosis patients in order to identify more efficient drug therapies [16]. The present study aimed to identify candidate biomarkers that can be used for the diagnosis of cystinosis and monitoring of its treatment through multi-omics studies. 

## 2. Results

The study included 33 patients with genetically confirmed cystinosis. The patient group included 14 patients (6 females and 8 males) with a median age of 1.04 years (6 months-10 years) that were diagnosed clinically and genetically during the study period, but were not receiving any treatment at the time of diagnosis (Table 1a). The treatment group included 19 patients (5 females and 14 males) with a median age of 3.5 years (7 months-17.3 years) that were previously diagnosed and were receiving treatment at the time the study began (Table 1b). The control group included 12 healthy females and 18 healthy males with a median age of 1.44 years (7 months to 10 years). The most common mutation observed in the patient and treatment groups was homozygous p.Glu227Glu [17], which is in agreement with our previous reports (Table 1) [18,19].

### 2.1. Metabolomic Results of Plasma Samples

The metabolite levels in plasma samples in all the study groups are presented in Appendix A. In total, 271 metabolites were annotated (Appendix A). Multivariate analysis was initially performed using the principal component analysis (PCA) scoring graphic and all data obtained from plasma metabolomics analysis in order to identify any systematic errors in the samples (Appendix A). Partial Least Squares-Discriminant Analysis (PLS-DA) was performed to evaluate differences between the groups and Variable Importance in Projection (VIP) graphs showing the most important metabolites leading to differences in data analysis were created (Figure 1). Binary comparisons between groups were performed using the PLS-DA score, and VIP and regression coefficient graphs were plotted (Appendix A). Univariate statistical methods were employed to identify significantly different metabolites. Among the 271 metabolites, we identified 46 metabolites that significantly differentiated the patient group from the treatment group and 74 metabolites that significantly differentiated the patient group from the healthy control group (Appendix A). In addition, targeted liquid chromatography-tandem mass spectrometry (LC-MS/MS) analysis showed that the taurine level was significantly higher in the treatment group than in the patient group, whereas the taurine, glutathione, and L-serine levels were significantly higher in the healthy control group than in the patient group (Appendix A). Binary comparisons, except for the comparison between patient and treatment groups, showed that the reliability (R^2^) and predictability (Q^2^) of the PLS-DA models were >0.5. Analysis Of VAriance testing of Cross Validated predictive residuals (CV-ANOVA) analysis of the PLS-DA models showed that the reliability of the models had high sensitivity (>90%) and significance (Fischer’s false classification probability *p* < 10^–7^) (Appendix A). These findings proved that the predictive power of the models was high. 

Moreover, the sensitivity and specificity of the candidate biomarkers were evaluated via Receiver Operating Characteristic (ROC) analysis for the patient group versus the treatment and healthy control groups separately. This analysis identified 15 and 14 metabolites that each differentiated the patient group from both the treatment and the healthy control groups, respectively (Figure 2).

### 2.2. Metabolomic Results of Urine Samples

This analysis included 61 urine samples, as follows: patient group: n = 14; treatment group: n = 17; and healthy control group: n = 30. Urine samples could not be appropriately obtained in two patients in the treatment group. Metabolic profiles were evaluated via a gas chromatography-mass spectrometry system (GC-MS), Liquid Chromatography quadrupole Time of Flight Mass Spectrometry (LC-qTOF-MS), and LC-MS/MS methods (Appendix A). In total, 376 metabolites were annotated (Appendix A). Initially, PCA analysis was performed to identify systematic errors (Appendix A), and the PLS-DA score plot and VIP graphs were created to evaluate differences between the groups and to identify the most important metabolites for in-group differentiation, respectively. Binary comparisons between the groups were also performed via PLS-DA and the PLS-DA score, and VIP and regression coefficient graphs were plotted (Appendix A).

Univariate statistical analysis showed that of the 376 metabolites, 57 differed significantly between the patient and treatment groups, and 199 differed significantly between the patient and healthy control groups (Appendix A). Moreover, targeted LC-MS/MS analysis showed that cystine differed significantly between the patient and treatment groups, and cystine, serine, methionine, and glutathione differed significantly between the patient and healthy control groups (Appendix A). R^2^ and Q^2^ of the PLS-DA models were >0.5 for all binary comparisons. The Fischer’s false classification probabilities of the CV-ANOVA analysis of the PLS-DA models were *p* < 10^–8^, showing that the reliability of the models had high sensitivity (>92%) (Appendix A). In addition, the sensitivity and specificity of the candidate biomarkers were evaluated via ROC analysis for the patient group versus the treatment and control groups separately (Figure 3). In all, 15 metabolites were identified that each differentiated the patient group from the treatment and control groups (Figure 3). 

### 2.3. Proteomic Results of Serum Samples

Shotgun proteomic analysis using LC-qTOF-MS was performed on serum samples, as follows: patient group: n = 12; treatment group: n = 19; and healthy control group: n = 29. In total, 65 proteins shared by the groups were identified, of which 1 protein differed significantly between the patient and treatment groups, 24 differed significantly between the patient and healthy control groups, and 16 differed significantly between the treatment and healthy control groups (Appendix A).

### 2.4. Pathway Analysis

Pathway analysis was performed using candidate biomarkers that were significantly different between groups. Highly relevant terms in the patient compared to the treatment group and in the patient group compared to the healthy control group are given in Appendix A, respectively. ABC transporters, central carbon metabolism in cancer, protein digestion and absorption, mineral absorption, the citrate cycle (TCA cycle), glyoxylate and dicarboxylate metabolism, and beta-alanine metabolism were the most significant pathways in the patient versus healthy control and treatment versus healthy control groups. The Krebs cycle; taurine and hypotaurine metabolism; alanine, aspartate, and glutamate metabolism; and glyoxylate and dicarboxylate metabolism were the most relevant pathways in the patient group versus the treatment group, and ascorbate and aldolate metabolism; taurine and hypotaurine metabolism; and the pentose phosphate pathway were the most significant relevant pathways in the treatment group versus in the healthy control group. 

## 3. Discussion

Early diagnosis of cystinosis is extremely critical in terms of growth, prevention of irreversible loss of organ functions, and ensuring a long life span in affected children [20]; therefore, it should be included in newborn screening programs. Currently used methods are ineligible for such purpose due to their many disadvantages; as such, there is an urgent need to identify new biomarkers that can be easily accessed and used in every center for cystinosis screening and monitoring the effectiveness of cystinosis treatment. The primary aim of biomarker research is to identify ≥1 metabolites that will facilitate differentiation of patients from healthy controls [21]. The present study for the first time, while identifying the molecular signatures of cystinosis utilizing a metabolomics approach in in vivo systems, identified the differences in metabolites present in plasma/urine samples that could be useful for differentiating cystinosis patients from healthy controls, thereby identifying candidate biomarkers that could be used in the future. 

The present study identified differences in metabolites between the three study groups, using PLS-DA score graphs that show how well the groups included in the study could be differentiated from each other, and the groups are expected to show the most discrete clustering possible. Given the fact that PLS-DA is a forced analysis method, its accuracy needs to be validated. As such, the present study calculated the reliability (R^2^) and predictability (Q^2^) values, and in binary comparisons, except for those between the patient and treatment groups, the R^2^ and Q^2^ of the PLS-DA models were >0.5, indicating that the methods used were valid [22]. Using the PLS-DA score graph of the plasma metabolomics samples, it was observed that the three study groups could be differentiated from each other on the basis of important metabolites. Moreover, when the plasma metabolomics data and the results of Fisher’s false classification analysis were evaluated together, it was determined that the patient group could be well differentiated from both the treatment group and control group with very high predictive power (>92%).

Appendix A clearly show that in the patient group the plasma levels of 3-indoleacetic acid, 3-phenyllactic acid, lyxose, glyceraldehyde, p-cresol, 4-trimethylammoniobutanoic acid, 5,8,11,14-octadecatetranoic acid, 4-isopropylbenzoic acid, PE(O-18:1(9Z)/0:0), and glycocholic acid were higher, while serine, taurine, alpha-D-glucosamine phosphate, glyceric acid, phosphoric acid, glutathione, homoserine, arachidic acid, glycolic acid, stearic acid, and N-acetyl-5-hydroxytryptamine metabolites were lower than in the control group. It is known that the end product of the breakdown of cysteine is taurine via cysteine sulfinic acid and hypotaurine intermediates [23]. Accordingly, the lower taurine level in the patients can be explained by the shortage of free cysteine, which is the well-known end-result in patients with cystinosis [1,5]. It is also known that alternative cysteine production is promoted via the serine cystathione metabolite [24]; therefore, a low serine pool might be related to its consumption to compensate for cellular free cysteine deficiency. Alternatively, serine is also converted to D-glycerate via hydroxypyruvate [24]. As the serine level decreases in patients with cystinosis, the rate of this pathway that leads to D-glycerate synthesis also decreases, which explains the reduced glycerate level in cystinosis patients. 

In healthy individuals, the source of cytosolic cysteine is cleavage of the intralysosomal cystine that is exported through the functional cystinosin transporter located on the lysosomal membrane. Cytosolic cysteine is involved in the synthesis of a tripeptide, γ-glutamyl-cysteinyl-glycine (glutathione), which plays an important role in the antioxidative defense mechanism [25,26]. In cystinosis patients, the production of glutathione is also interrupted due to a shortage of free cytosolic cysteine [25,26], which explains the low glutathione levels observed in the present study. Furthermore, N-acetyl-5-hydroxytryptamine functions as an antioxidant in the central nervous system and can cross the blood-brain barrier [27,28]. In the present study, the level of this metabolite also decreased along with the glutathione level in the cystinosis patients, suggesting that multiple antioxidant mechanisms are affected in cystinosis patients.

Increases in such metabolites as lyxose and 3-indolacetic acid in cystinosis patients are related to the metabolic pathways of aromatic amino acids (i.e., phenylalanine, tyrosine, and tryptophan). Lyxose is an intermediate released during the physiological fermentation process of bacteria in the intestinal flora. It was reported that the plasma lyxose concentration in hypertensive individuals increases along with deterioration of the physiological intestinal microbiota [29,30]. Tryptophan and 3-indolacetic acid play a role in intestinal immune homeostasis and can be altered via disruption of the composition of intestinal microbiota [31]. Whereas phenylacetate is an end-product of the phenyl alanine degradation pathway [32], p-cresol is an end-product of degradation of tyrosine by intestinal bacteria [33]. These findings indicate that the intestinal microbiota and/or the metabolic activity of the bacteria in the microbiota are altered in cystinosis patients; therefore, it can be concluded that in cystinosis patients the metabolism of aromatic amino acids is negatively affected in many ways and might play a role in the pathophysiology of the disease. The present study’s ROC analysis identified 10 significant plasma metabolites (L-serine, taurine, lyxose, 4-trimethylammoniobutanoic acid, orotic acid, glutathione, PE(O-18:1(9Z)/0:0), 2-hydroxyphenylacetic acid, acetyl-N-formyl-5-methoxyquinuramine, and 3-indoxyl sulfate) that can differentiate cystinosis patients from healthy controls and could be considered to have biomarker potential.

The present study also evaluated urine metabolomics, which, as shown in Appendix A, differed between the study groups. Based on these differences, the patient and control groups were differentiated from each other with very high predictive power (92%). In addition, the patient and treatment groups, and the treatment and control groups were differentiated with predictive power of 100%, indicating that urine metabolomics could be used for the diagnosis and follow-up of cystinosis (Appendix A). 

As presented in Appendix A, eight metabolites (allo-inositol, talose, glucose, 2-hydroxybutyric acid, cystine, pyruvic acid, valine, and phenylalanine) that were significantly higher and nine metabolites (N-acetyl-L-glutamic acid, 3-aminopropionitrile, ribitol, hydroquinone, glucuronic acid, 3-phosphoglycerate, xanthine, creatinine, and 5-aminovaleric acid) that were significantly lower in the cystinosis patients differentiated them from the healthy controls. The detection of metabolites in urine (glucose, cystine, valine, and phenylalanine) in cystinosis patients that are normally not present in the urine of healthy individuals could be associated with tubular damage (i.e., Fanconi syndrome) that develops in cystinosis patients [34]. 

2-hydroxybutyric acid is produced in mammalian tissues that synthesize glutathione. Under the stress conditions of cystinosis, as sources of L-cysteine for glutathione synthesis are limited, metabolism is directed towards alternative sources; cystathione is such an alternative. As cystathione is converted to cysteine, 2-hydroxybutyrate is released as a by-product [35,36]. The increased urinary 2-hydroxybutyric acid level in the present study’s patients suggests that glutathione synthesis via this salvage pathway occurs in cystinosis patients. 

L-Glutamate is a precursor of both glutathione and N-acetylglutamic acid. In mammals, N-acetylglutamic acid is the allosteric activator of mitochondrial carbamoyl phosphate synthetase I, the enzyme that catalyzes the first step of the urea cycle. The rate of glutathione synthesis is dependent on the concentration of intracytoplasmic cysteine [37], which is low when compared to the other two amino acids present in the glutathione structure [38] that drives carbamoyl phosphate synthesis and results in a decrease in urinary excretion of N-acetylglutamate [39]. The high orotic acid level observed in the plasma metabolomics of the present study’s cystinosis patients also supports this explanation. Indeed, it has been demonstrated that a higher level of carbamoyl phosphate than is needed for the urea cycle is directed to the synthesis of orotic acid, orotidine, and uracil [40].

Comparison of the present study’s patient and control groups based on VIP and ROC curve analysis of urine metabolites showed that ribitol and an increase in cystine metabolites were the best candidate biomarkers. Comparison of the treatment and control groups based on VIP and ROC curve analysis of urine metabolite, hypotaurine, phosphatidylethanolamine, N-acetyl-d-mannosamine, 3-indolacetic acid, p-cresol, 5-aminovaleric acid, talose, phenylethylamine, and 2-hydroxybutyric acid levels showed that the levels of hydroquinone, citramalic acid, oleic acid, 3-aminopropionitrile, aconitic acid, pantothenic acid, ribitol, fucose, ribose, and creatinine were lower in the treatment group. The most striking finding was that the urinary cystine level, which was higher in the patient group than in the control group, was not detected in urine in the treatment group—an expected finding considering the therapeutic action of cysteamine. 

Pathway analysis showed that the most significantly altered pathways with the highest variation between the patient and control groups were the ascorbate–aldarate metabolism, taurine–hypotaurine metabolism, and pentose phosphate pathways. When both plasma and urine metabolomics data were integrated, the most affected metabolic pathways in the cystinosis patients were related to carbon metabolism and ATP-binding cassette (ABC group) transporters. ABC transporters are a superfamily of integral membrane proteins commonly found in organisms that are responsible for ATP-assisted translocation of many substrates across membranes. The best-characterized ABC transporter is P-glycoprotein (P-gp), which was initially observed to be overexpressed in drug-resistant tumor cells [38]. Later, it was shown that P-gp is an efflux pump, an important component of barrier tissues, and a factor that plays an important role in the excretion of drugs. Expression of P-gp in the kidneys is restricted to the apical membrane of the proximal tubule, consistent with its urinary excretory function. Indeed, the present study’s pathway analysis findings are in agreement with the findings reported in mice with defective P-gp, in which apical proximal tubular dysfunction resembling human Fanconi syndrome was observed [41]. Taurine and hypotaurine metabolism, sulfur metabolism, the pentose phosphate pathway, the citric acid cycle and glutathione metabolism, mineral absorption, aminoacyl-tRNA biosynthesis, glyoxylate and dicarboxylate metabolism, cholesterol metabolism, ascorbate and aldarate metabolism, unsaturated fatty acids biosynthesis, pantothenate and CoA biosynthesis, pentose and glucuronate transformations, nicotinate, and nicotinamide metabolism, as well as protein and amino acid metabolism were significantly altered in the present study’s cystinosis patients. The most severely affected amino acids in the patients were phenylalanine, tyrosine, tryptophan, arginine, valine, leucine, isoleucine, proline, serine, threonine, cysteine, aspartate, glutamate, methionine, alanine, and glycine. These findings support the notion that the pathogenesis of cystinosis is complex and beyond intralysosomal cystine accumulation [2]. 

The present study’s findings (i) identify new candidate biomarkers that can be used for the diagnosis of cystinosis; (ii) provide a basis for the development of alternative diagnostic methods that can overcome the disadvantages of current diagnostic methods; (iii) provide guidance for future studies on the development of more effective cystinosis treatment options that can be partially controlled with cysteamine therapy; and (iv) provide knowledge that can guide future research on the diagnosis, follow-up, and treatment of other rare diseases.

## 4. Materials and Methods

### 4.1. Definitions

Cystinosis was diagnosed based on the observation of corneal cystine crystals via eye examination and/or an elevated leukocyte cystine level (0.2 nmol half-cystine mg^–1^ protein), and/or identification of bi-allelic *CTNS* mutations. Fanconi syndrome was diagnosed based on standard clinical, laboratory, and biologic criteria, including growth retardation, polyuria-polydipsia, electrolyte imbalance, dehydration, and/or rickets [2].

### 4.2. Patients and Controls

The study included 14 newly diagnosed cystinosis patients not under treatment at the time of inclusion in the study (patient group), 19 previously diagnosed cystinosis patients that were under treatment (treatment group), and 30 sex- and age-matched healthy children (control group) (Figure 4). All cases of cystinosis were genetically confirmed (Table 1). Plasma, serum, and urine samples were collected from the study groups and were stored at –80 °C until analyzed. The study protocol was approved by the Hacettepe University Ethics Committee (no. GO 18/237-30) and was performed in accordance with the Declaration of Helsinki. Written informed consent was obtained from the parents of all the participants.

### 4.3. Genomic Studies

Genetic testing of the patients was performed at the Hacettepe University Nephrogenetics Laboratory. Genomic DNA was extracted from whole blood using the Invitrogen Pure Link Genomic DNA Mini Kit^TM^ (Thermo Fisher Scientific, Waltham, CA, USA), according to the manufacturer’s instructions. All exons of the *CTNS* gene, together with their adjacent intron junctions were analyzed via direct sequencing using BigDye™ Terminator v3.1 Cycle Sequencing Kit (Thermo Fisher Scientific, Waltham, CA, USA) and an ABI3130 genetic analyzer (Applied Biosystems, Foster City, CA, USA). The National Center for Biotechnology Information transcript variant 1 of CTNS (NM_001031681.2), corresponding to ENSEMBL transcript ENST00000381870.7, was used as the reference sequence. 

### 4.4. Omics Analysis

Untargeted proteomic and metabolomic analysis was performed in all the study groups (Appendix A). Moreover, targeted pathway analysis for sulfur metabolism, including cystine, cysteine, serine, glutathione, and taurine, was performed via liquid chromatography-tandem mass spectrometry (LC-MS/MS) (Figure 4). The methods used are briefly summarized below and the details are given in Appendix A. 

### 4.5. Metabolomic Analysis 

#### 4.5.1. Sample Preparation

An aliquot of 200 µL plasma or 100 µL urine sample from each patient was separately transferred into the Eppendorf tubes. Before extraction of the metabolites, the urine samples were treated with 100 µL of urease enzyme at 37°C for 30 min. 

The plasma and urine metabolites were extracted using methanol:water (9:1, *v/v*) mixture containing 1 µg/mL myristic acid as internal standard. Then, the extracts were centrifuged at 15,000 rpm for 5 min (at +4 °C) and 200 µL aliquots of each sample supernatant were transferred to three Eppendorf tubes and completely dried by a vacuum centrifuge for GC-MS, LC-qTOF-MS, and LC-MS/MS-based metabolomics analysis. 

The extracted samples were analyzed using a gas chromatography-mass spectrometry system (GC-MS-QP-2010 Ultra system, Shimadzu, Kyoto, Japan) and LC-qTOF-MS (Agilent, California, USA). The representative chromatograms of GC-MS and LC-qTOF-MS based metabolomics analysis for plasma and urine samples are given in Appendix A, respectively. The detailed protocols for GC-MS and LC-qTOF-MS analysis are given in Appendix A.

#### 4.5.2. Targeted Pathway Analysis for Sulfur Metabolism

The LC-MS/MS (Shimadzu 8030, Kyoto, Japan) was used for targeted analysis of the sulfur metabolism pathway metabolites, including cystine, cysteine, serine, glutathione, and taurine from plasma and urine samples. The detailed protocols (Appendix A) and MRM parameters and gradient elution program (Appendix A) for LC-MS/MS analysis are given in Appendix A.

#### 4.5.3. Proteomic Analysis 

Proteomic profiling in all the study groups was performed via shotgun proteomic analysis after the depletion of the 12 most abundant proteins using immuno-affinity kits from a 10 µL plasma sample. Then, the proteins were reduced and alkylated, and trypsin-based digestion was carried out to generate peptides using LC-qTOF-MS (Agilent, California, USA) analysis (Appendix A). The detailed protocol is given in Appendix A.

### 4.6. Data Processing 

Data deconvolution and/or analysis for metabolomic and proteomic studies were conducted utilizing several software packages. Details are given in Appendix A. Briefly, PCA and PLS-DA) were performed. The VIP values were estimated to identify the most critical metabolites for stratification of the groups, and regression coefficients were exploited to illustrate the effects of the metabolites on each group. ROC curve analysis with support vector machines (SVM) was used to identify potential diagnostic references and to evaluate their performance. Joint pathway analysis was performed for significantly different metabolites and proteins. 

### 4.7. Statistical Analysis

All statistical analysis of MS data sets were carried out in R Programming Language v.4.1 with base packages (www.R-project.org/ (accessed on 1 September 2022)). Data were analyzed for general tendencies and dispersion characteristics. Mean ± SD and median (min-max) were calculated. The Shapiro–Wilk normality test was used to determine if data were normally distributed in the 95% CI. Data with normal distribution were compared via the two-tailed Student’s *t*-test. On the other hand, a comparison of non-normally distributed data was performed using the Mann-Whitney U test. Paired samples in the dataset were processed using the paired *t*-test or the Wilcoxon signed rank test, according to the normality of data distribution. The results were evaluated by *p* values in the 95% CI. A *p* value <0.05 was considered statistically significant.

## Figures and Tables

**Figure 1 ijms-24-02603-f001:**
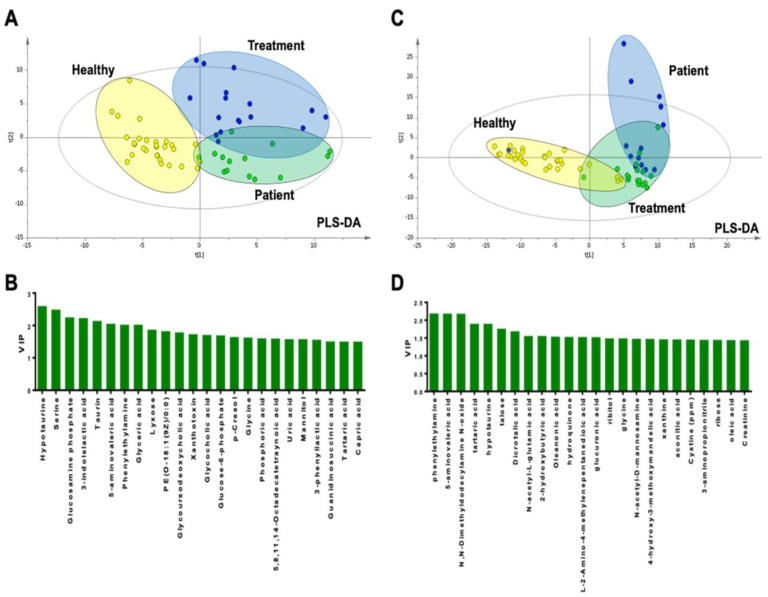
Multivariate analysis of data obtained from plasma (left panel) and urine (right panel) metabolomics. (**A**,**C**): PLS-DA score graphs; (**B**,**D**): VIP graphs showing the most important metabolites leading to differences in data analysis.

**Figure 2 ijms-24-02603-f002:**
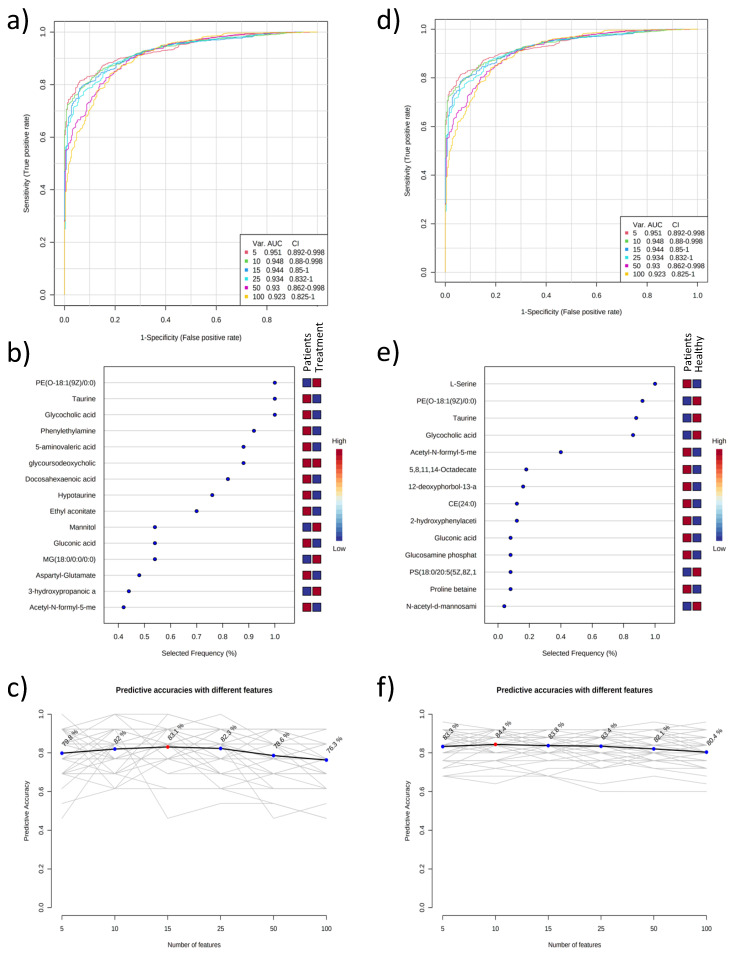
Multivariate ROC curve-based exploration graphs for plasma metabolomics analysis. Patient group and treatment group (**a**–**c**); patient group and healthy control group (**d**–**f**) (**a**,**d**); selectivity and sensitivity graphs (**b**,**e**); the most significant metabolites on the estimation of the method (**c**,**f**); effects of different numbers of metabolites on the estimation of the method.

**Figure 3 ijms-24-02603-f003:**
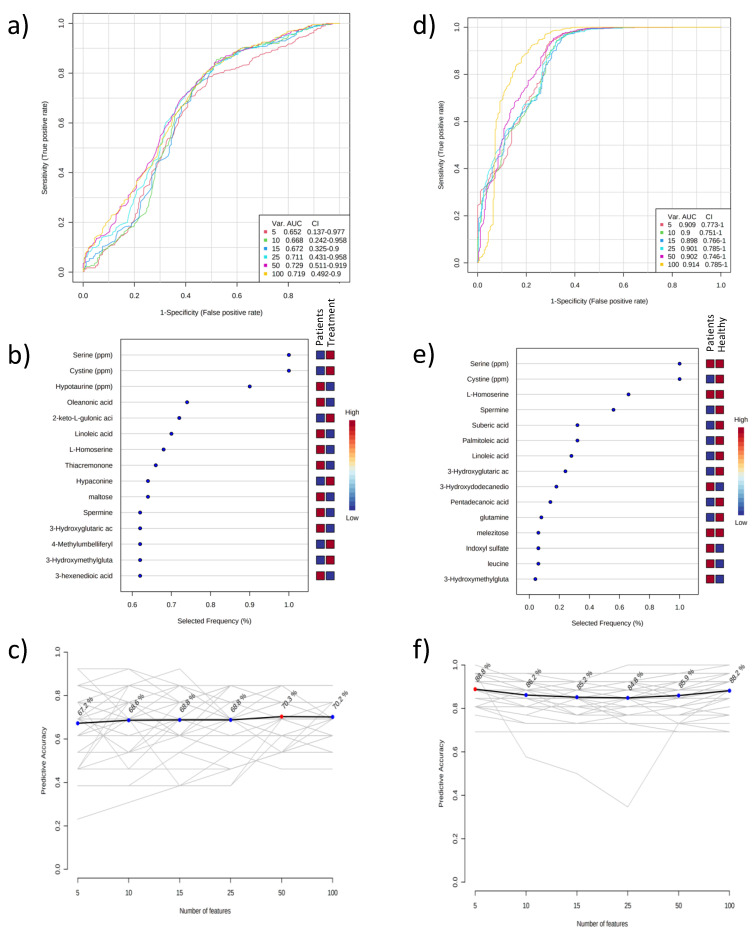
Multivariate ROC curve-based exploration graphs for urine metabolomics analysis. Patient group and treatment group (**a**–**c**); patient group and healthy control group (**d**–**f**) (**a**,**d**); selectivity and sensitivity graphs (**b**,**e**); the 15 most significant metabolites on the estimation of the method (**c**,**f**); effects of different numbers of metabolites on the estimation of the method.

**Figure 4 ijms-24-02603-f004:**
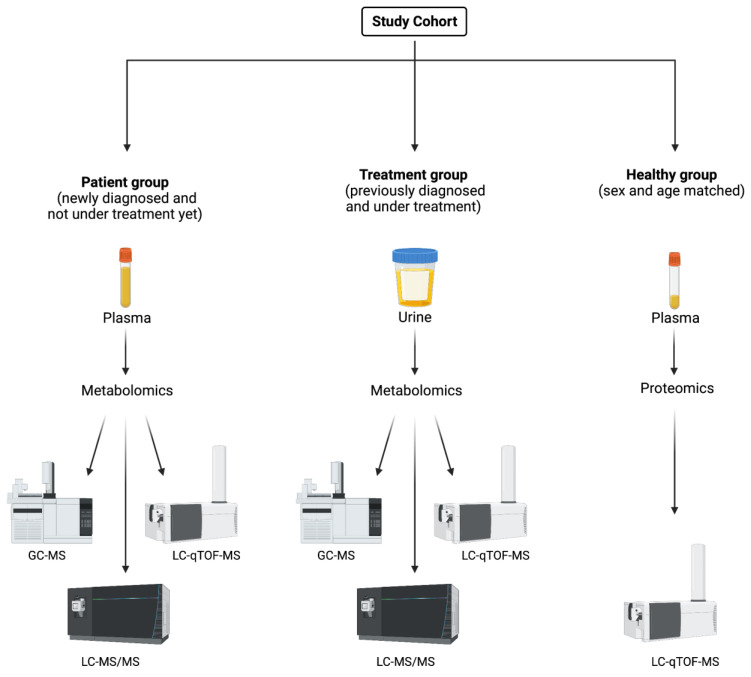
Study cohort and multi-omics analyses were performed on multiple biological specimens using multiple analytical techniques.

**Table 1 ijms-24-02603-t001:** Gender, age at sampling, and mutation detected in the patient and treatment groups.

**(a) Patient Group**			
**No**	**Gender**	**Mutation**	**Age at Sampling (Years)**
1	Male	c.681G>A (p.Glu227Glu) (Hom)	1.16
2	Male	c.681G>A (p.Glu227Glu) (Hom)	6.97
3	Male	c.681G>A (p.Glu227Glu) (Hom)	1.54
4	Male	c.18_21 del (p.Thr7Phefs*7) (Hom)	0.91
5	Male	c.1015G>A (p.Gly339Arg) (Hom)	1.68
6	Male	c.18_21 del (p.Thr7Phefs*7) (Hom)	0.76
7	Female	c.681G>A (p.Glu227Glu) (Hom)	10
8	Female	c. 691C>T (Gln231*) (Hom)	0.74
9	Female	c.291_294del (p. Thr98Phefs*19) (Hom)	0.6
10	Female	c.681G>A (p.Glu227Glu) (Het)/c.1054 C>T(p.Gln352*) (Het)	2.11
11	Female	c.323delA (p.Gln108Argfs*10) (Hom)	2.23
12	Male	c.18_21 del (p.Thr7Phefs*7) (Hom)	0.6
13	Female	c.141–22 A>G (Hom)	0.8
14	Male	c.681G>A (p.Glu227Glu) (Hom)	0.8
**(b) Treatment Group**			
**No.**	**Gender**	**Mutation**	**Age at Sampling (Years)**
1	Male	c.681G>A (p.Glu227Glu) (Hom)	3.16
2	Male	c.681G>A (p.Glu227Glu) (Hom)	8.46
3	Male	c.681G>A (p.Glu227Glu) (Hom)	2.95
4	Male	c.18_21 del (p.Thr7Phefs*7) (Hom)	2.16
5	Male	c.1015G>A (p.Gly339Arg) (Hom)	2.93
6	Male	c.18_21 del (p.Thr7Phefs*7) (Hom)	1.70
7	Female	c.681G>A (p.Glu227Glu) (Hom)	10.63
8	Female	c.681G>A (p.Glu227Glu) (Het)/c.1054 C>T(p.Gln352*) (Het)	2.27
9	Male	c.18_21 del (p.Thr7Phefs*7) (Hom)	0.58
10	Female	c.291_294del (p. Thr98Phefs*19) (Hom)	1.08
11	Female	c.681G>A (p.Glu227Glu) (Hom)	10.26
12	Female	c.681G>A (p.Glu227Glu) (Hom)	1.84
13	Male	c.681G>A (p.Glu227Glu) (Hom)	17.3
14	Male	c.681G>A (p.Glu227Glu) (Hom)	8.80
15	Male	c.1015G>A (p.Gly339Arg) (Hom)	13.94
16	Male	c.1015G>A (p.Gly339Arg) (Hom)	6.48
17	Male	c.18_21 del (p.Thr7Phefs*7) (Hom)	3.50
18	Male	c.323delA (p.Gln108Argfs*10) (Hom)	7.97
19	Male	c.141–22A>G (het)/c.681G>A (p.Glu227Glu) (Het)	10.61

Variations and predicted amino acid changes in parantheses were named according to the guidelines of the Human Genome Variation Society; transcript no: NM_001031681.2.* indicates stop codon. Hom, homozygous; Het, heterozygous.

## Data Availability

Data is contained within the article and Appendix A.

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
