# Peer review of "Metabolomic Analyses to Identify Candidate Biomarkers of Cystinosis"

_ijms, 2023, doi:10.3390/ijms24032603_

Round 1

Reviewer 1 Report

My only concerns relate to the introduction to manuscript, in two places:

1. The authors state " For their survival such patients must take almost 60 different drugs to replace urinary losses. Nevertheless, use of these drugs cannot prevent loss of glomerular function, which eventually leads to chronic renal failure necessitating renal replacement therapy (dialysis or renal transplantation) for survival" page 2 lines 50-53.

Points - "Almost 60" is not accurate for the vast majority of patients.  Also, the main factor that prevents renal failure is cysteamine, not the drugs used to replace urinary losses.

2. Page 2 line 70/71 states "In addition, this method is not practical 70 in clinics, and is not widely available."

Point - Assays for cystine are in fact both practical in a clinical setting and are widely available and widely used in modern healthcare settings.

Author Response

Response-1: Thank you for your valuable comment. This sentence was revised as “For their survival such patients should take many different drugs to replace urinary losses. Along with this supportive treatment, cysteamine is the specific therapy. Early initiation of cysteamine has been shown to delay progression to chronic kidney failure.” Spell check was also performed, as suggested. 

Response-2: Thank you for your valuable comment. This sentence was completely removed as other sentences already point to the disadvantages of measurements of leukocyte cystine levels.

Reviewer 2 Report

Nice paper.  They used well-established metabolomic analysis to see if any biomarkers can be used to fingerprint cystinosis.  It is difficult to really know for sure whether found biomarkers will hold up over time.  At least the taurine observations have some biochemical rational related to cysteine metabolism.

Author Response

Response 1: Thank you for your supportive and valuable comment.
